# Inertial delay of self-propelled particles

Christian Scholz [1], Soudeh Jahanshahi[1], Anton Ldov[1] & Hartmut Löwen[1]

The motion of self-propelled massive particles through a gaseous medium is dominated by inertial effects. Examples include vibrated granulates, activated complex plasmas and flying insects. However, inertia is usually neglected in standard models. Here, we experimentally demonstrate the significance of inertia on macroscopic self-propelled particles. We observe a distinct inertial delay between orientation and velocity of particles, originating from the finite relaxation times in the system. This effect is fully explained by an underdamped generalisation of the Langevin model of active Brownian motion. In stark contrast to passive systems, the inertial delay profoundly influences the long-time dynamics and enables new fundamental strategies for controlling self-propulsion in active matter.

[1] Institut für Theoretische Physik II: Weiche Materie, Heinrich-Heine-Universität Düsseldorf, 40225 Düsseldorf, Germany. Correspondence and requests for materials should be addressed to C.S. (email: christian.scholz@hhu.de) or to H.L. (email: hlowen@thphy.uni-duesseldorf.de)

N ewton's first law states that because of inertia, a massive object resists any change of momentum. Before this groundbreaking idea, the dominant theory of motion was based on Aristotelian physics, which posits that objects come to rest unless propelled by a driving force. In retrospect, this perception is unsurprising, as the motions of everyday objects are influenced significantly by friction. In microscopic systems such as colloids, inertial forces are completely overwhelmed by viscous friction. In fact, in the absence of inertia, particles cannot move by reciprocal shape deformations due to kinematic reversibility. Biological organisms such as bacteria must therefore self-propel by implementing non-reciprocal motion[1].

However, any finitely massive object performs ballistic motion, even if only on minuscule time and length scales. For example, colloidal particles undertake ballistic motion below 1 Å for approximately 100 ns. Experimental verification of this motion requires high accuracy measurements and has been achieved only for passive colloids[2–4]. In contrast, for macroscopic self-propelled particles, such as animals and robots, the magnitude of inertial forces can be comparable to that of the propulsion forces and influence the dynamics on large time scales.

A particularly simple example of a macroscopic self-propelled particle is a minimalistic robot called a vibrobot, which converts vibrational energy into directed motion using its tilted elastic legs[5]. Collectives of such particles exhibit novel non-equilibrium dynamics[6–9], self-organisation[10], clustering[5,11] and swarming[12–15]. Along with animals[16], artificial and biological microswimmers[17–19], vibrobots belong to the class of active soft matter.

Here, we demonstrate that the inertia of self-propelled particles causes a significant delay between their orientation and velocity and increases the long-time diffusion coefficient through persistent correlations in the underdamped rotational motion. Standard models, such as the Vicsek-model[20] and active Brownian motion[21] cannot explain this behaviour as they neglect inertia. Instead, the dynamics can be understood in terms of underdamped Langevin equations with a self-propulsion term that couples the rotational and translational degrees of freedom. Using the mean squared displacements (MSDs) and velocity distributions, fitted by numerical and analytical results, we extract a unique set of parameters for the model. We derive analytic solutions for the short- and long-time behaviour of the MSD and prove that the long-time diffusion coefficient explicitly depends on the moment of inertia.

## Results

**Experimental observation of inertial effects.** Our experimental particles are 3D-printed vibrobots driven by sinusoidal vibrations from an electromagnetic shaker. To investigate a wide range of parameter combinations, we varied the leg inclination, mass and moment of inertia of the particles (see Fig. 1a–d). The excitation frequency and amplitude were fixed to $f = 80$ Hz and $A = 66$ μm, respectively, which ensures stable quasi-two-dimensional motion of the particles.

The mechanism is illustrated in Fig. 1e and Supplementary Movie 1. The vibrobots move by a ratcheting mechanism driven by repeated collisions of their tilted elastic legs on the vibrating surface. Their propulsion velocity depends on the excitation frequency, amplitude, leg inclination and material properties such as the elasticity and friction coefficients[5,22–24]. Long-time random motions are induced by microscopic surface inhomogeneities and (under sufficiently strong driving) a bouncing ball instability[24], that causes the particle's legs to jump asynchronously and perform a tiny but very irregular precession, which in turn leads to random reorientations of the particle. Thereby, the vibrobot motion is considered as a macroscopic realisation of active

Brownian motion[12,13,25,26]. Figure 1f shows three representative trajectories of particles with different average propulsion velocities (see also Supplementary Movie 2). The persistence length is noticeably shorter for slower particles than for faster particles, as generally expected for self-propelled particles[19].

However, the significance of inertial forces is an important difference between motile granulates and microswimmers[11,27]. Massive particles do not move instantaneously, but accelerate from rest when the vibration is started. The time dependence of the initial velocity (averaged over up to 165 runs per particle) is shown in Fig. 2a. The particles noticeably accelerated up to the steady state on a time scale of $10^{-1}$ s, one order of magnitude larger than the inverse excitation frequency and the relaxation time of the shaker. When perturbed by an external force, vibrationally driven particles approach their steady state on a similar time scale[10]. The relaxation process is well fitted by an exponential function, as expected for inertial relaxation. Inertia also influences the dynamical behaviour of the particles' orientation relative to their velocity. The orientation (red arrows in Fig. 2b) systematically deviates from the movement direction (black arrows in Fig. 2b). Particularly, during sharp turns the orientation deviates towards the centre of the curve, whereas the velocity is obviously tangential to the trajectory. We compare the angle of orientation $\phi$ to the angle of velocity $\Theta = \text{atan2}(\dot{Y}, \dot{X})$ in Fig. 2b and find that $\Theta$ systematically pursues $\phi$ with an inertial delay of order $10^{-1}$ s. A slow-motion recording of one particle in Supplementary Movie 3 illustrates the dynamic delay between motion and orientation. The particle quickly reorients, but its previous direction is retained by inertia. Consequently, the particle drifts around the corner, mimicking the well-known intentional oversteering of racing cars.

**Underdamped Langevin model.** Despite the complex non-linear dynamics of the vibrobots[5,23,24,28,29], our observations can be fully described by a generalised active Brownian motion model with explicit inertial forces. The dynamics are characterised by the centre-of-mass position $\mathbf{R}(t) = (X(t), Y(t))$ and the orientation $\mathbf{n}(t) = (\cos\phi(t), \sin\phi(t))$, where $\phi(t)$ defines the direction of the propulsion force. The coupled equations of motion for $\mathbf{R}(t)$ and $\phi(t)$, describing the force balance between the inertial, viscous and random forces, are given by

$$M\ddot{\mathbf{R}}(t) + \xi\dot{\mathbf{R}}(t) = \xi V_p \mathbf{n}(t) + \xi\sqrt{2D}\mathbf{f}_{st}(t), \qquad (1)$$

$$J\ddot{\phi}(t) + \xi_r\dot{\phi}(t) = \tau_0 + \xi_r\sqrt{2D_r}\tau_{st}(t). \qquad (2)$$

Here, $M$ and $J$ are the mass and moment of inertia, respectively, and $\xi$ and $\xi_r$ denote the translational and rotational friction coefficients. The translational and rotational Brownian fluctuations are quantified by their respective short-time diffusion coefficients $D$ and $D_r$. The random forces $\mathbf{f}_{st}(t)$ and torque $\tau_{st}(t)$ are white noise terms with zero mean and correlation functions $\langle \mathbf{f}_{st}(t) \otimes \mathbf{f}_{st}(t') \rangle = \delta(t - t')\mathbf{1}$ and $\langle \tau_{st}(t)\tau_{st}(t') \rangle = \delta(t - t')$, respectively, where $\langle \cdots \rangle$ denotes the ensemble average and $\mathbf{1}$ is the unit matrix. Owing to the strong non-equilibrium nature of the system, the diffusion and damping constants are not related by the Stokes−Einstein relation[30]. Moreover, as typical particles are not perfectly symmetrical, they tend to perform circular motions on intermediate time scales. To capture this behaviour, we applied an external torque $\tau_0$ that induces circular movement with average velocity $\omega = \tau_0/\xi_r$[31,32]. Similar models applied in the literature have typically neglected the moment of inertia or have only been solved numerically[11,33–37]. The motion of a particle governed by Eqs. (1) and (2) is determined by different time scales given by the friction rates $\xi/M = \tau^{-1}$ and $\xi_r/J = \tau_r^{-1}$, the

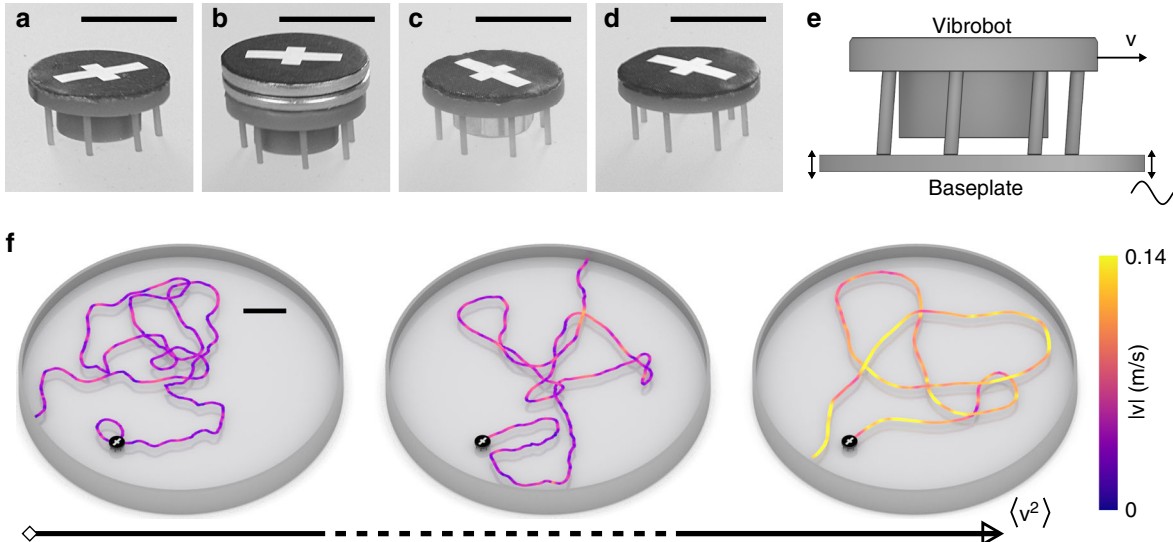

**Fig. 1** 3D-printed particles, setup and trajectories. **a** Generic particle. **b** Carrier particle with an additional outer mass. **c** Tug particle with an additional central mass. **d** Ring particle without a central core. Scale bars represent 10 mm. **e** Illustration of the mechanism with a generic particle on a vibrating plate. **f** Three exemplary trajectories with increasing average particle velocities. Particle images mark the starting point of each trajectory. The trajectory colour indicates the magnitude of the velocity. Scale bar represents 40 mm

rotational diffusion rate $D_r$, the angular frequency $\omega$ and the crossover times $2D/V_p^2$ and $2D_r/\tau_0^2$. In the limit of vanishing $M$ and $J$ the model is equivalent to the well-known active Brownian motion formulation[21].

The trajectories obtained by numerically integrating the Langevin model compare well with the experimental observations. As show by the representative trajectory in Fig. 2d, e, the model reproduces the delay between the orientation and velocity, when the friction is sufficiently weaker than the inertia. The model can be analytically solved by averaging and integration. The orientational correlation

$$\langle \mathbf{n}(t) \cdot \mathbf{n}(0) \rangle_\mathrm{T} = \cos(\omega t) e^{-D_r \left( t - \tau_r \left( 1 - e^{-t/\tau_r} \right) \right)}, \tag{3}$$

where $\langle \cdots \rangle_\mathrm{T}$ is the time average, quantifies the temporal evolution of the active noise term. The periodic cosine term results from the external torque and captures the induced circular motion. The rotational noise, quantified by $D_r$, decorrelates the orientation on long-time scales. This decorrelation is described by the exponential term in Eq. (3). The double exponential reflects the additional orientation correlation on short time scales imposed by the inertial damping rate $\tau_r^{-1}$. Consequently, the particle dynamics non-trivially depend on the orientation, even in the short- and long-time limits. In the short-time limit the MSD is given by

$$\left\langle (\mathbf{R}(t) - \mathbf{R}_0)^2 \right\rangle = \left\langle \dot{\mathbf{R}}^2 \right\rangle t^2 \tag{4}$$

with

$$\left\langle \dot{\mathbf{R}}^2 \right\rangle = 2D/\tau + \mathfrak{f}(\mathfrak{D}_0, \mathfrak{D}_1, \mathfrak{D}_2) V_p^2. \tag{5}$$

The first term is the equilibrium solution for a passive particle, and the second term arises from the active motion term. The latter is proportional to $V_p^2$, i.e. the kinetic energy injected by the propulsion. This contribution is quantified by the ratio of competing time scales, i.e. the dimensionless delay numbers

$$\mathfrak{D}_0 = D_r \tau_r, \quad \mathfrak{D}_1 = \omega \tau_r, \quad \mathfrak{D}_2 = \tau_r/\tau, \tag{6}$$

through the function

$$\mathfrak{f}(\mathfrak{D}_0, \mathfrak{D}_1, \mathfrak{D}_2) = \mathfrak{D}_2 e^{\mathfrak{D}_0} \mathrm{Re}\left[ \mathfrak{D}_0^{-(\mathfrak{D}_0 - i\mathfrak{D}_1 + \mathfrak{D}_2)} \right. $$
$$\left. \times \gamma(\mathfrak{D}_0 - i\mathfrak{D}_1 + \mathfrak{D}_2, \mathfrak{D}_0) \right], \tag{7}$$

where Re denotes the real part and $\gamma$ is the lower incomplete gamma function. The long-time behaviour of the motion is diffusive, with the long-time diffusion coefficient

$$D_\mathrm{L} = D + \frac{V_p^2}{2} \mathfrak{t}(\tau_r, \mathfrak{D}_0, \mathfrak{D}_1). \tag{8}$$

In Eq. (8), the first term is the passive diffusion coefficient and the second term represents the contribution from the driving force with persistence time given by

$$\mathfrak{t}(\tau_r, \mathfrak{D}_0, \mathfrak{D}_1) = \tau_r e^{\mathfrak{D}_0} \mathrm{Re}\left[ \mathfrak{D}_0^{-(\mathfrak{D}_0 - i\mathfrak{D}_1)} \gamma(\mathfrak{D}_0 - i\mathfrak{D}_1, \mathfrak{D}_0) \right]. \tag{9}$$

Equation (8) is similar to the active Brownian motion model, where the persistence time $1/D_r$ is replaced by Eq. (9). The long-time diffusion coefficient is therefore a function of the inertial correlations introduced by $J$ through $\mathfrak{D}_0$. This starkly contrasts with passive Brownian motion, which assumes an inertia-independent diffusion coefficient.

**Comparison between model and measurement.** Equations (5) and (8) depend non-trivially on six independent parameters. They are determined by fitting the MSD given by Eq. (4) and the linear and absolute velocity distributions, obtained by numerically solving Eqs. (1) and (2), to the measurements. The measurements and fitting curves for the four different particle types are summarised in Fig. 3. The angular MSDs in Fig. 3a–d show a ballistic short-time regime and a diffusive long-time regime (dashed lines) from which we can determine $\tau_r$ and $D_r$, respectively. The $\dot{\phi}$-distribution in Fig. 3e–h is a shifted Gaussian. The minor deviations at small velocities are caused by the finite tracking accuracy. The first moment of this distribution gives the mean angular velocity $\omega$. The parameters $\tau$, $D$ and $V_p$ are extracted from the linear velocity distributions $P(v_\mathrm{lin}) = P(\dot{X}) = P(\dot{Y})$

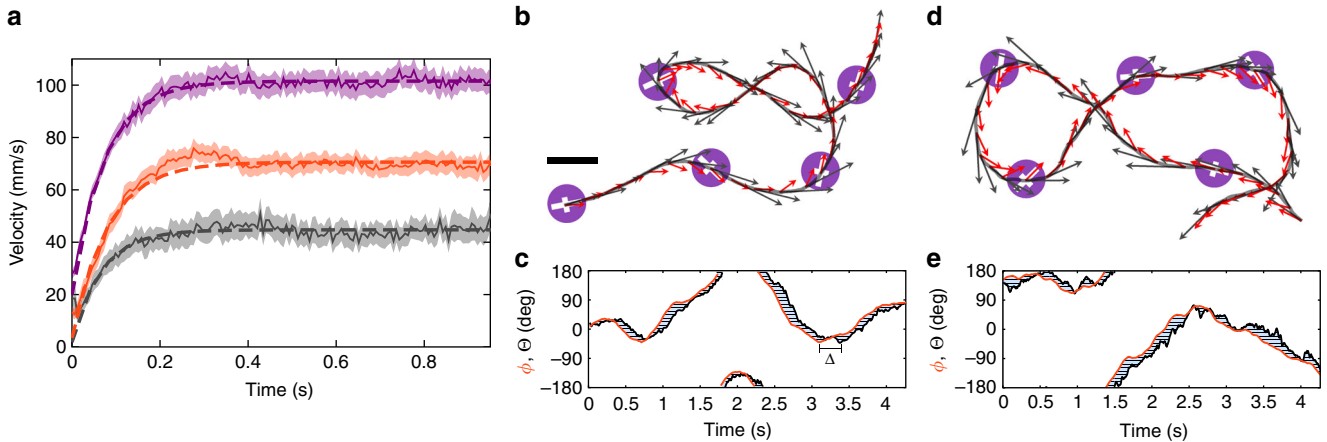

**Fig. 2** Inertial delay in particle trajectories. **a** Time-dependence of the average particle velocity starting from rest at $t_0$ for three particles with different leg inclinations 2° (grey), 4° (red) and 6° (violet). **b** Measured particle trajectory showing the direction (black arrows) and orientation (red arrows) of a particle. Scale bar represents 20 mm. **c** The measured orientation curve $\phi(t)$ (red) lags the velocity direction curve $\Theta(t)$ (black) by an inertial delay $\Delta$. **d** Corresponding simulated trajectory with velocity direction (black arrows) and orientation (red arrows). The model parameters are $\xi/M = 6.46 \, \mathrm{s^{-1}}$, $\xi_r/J = 5.4 \, \mathrm{s^{-1}}$, $D = 8 \times 10^{-5} \, \mathrm{m^2 \, s^{-1}}$, $D_r = 2.59 \, \mathrm{s^{-1}}$, $V_p = 0.092 \, \mathrm{ms^{-1}}$, $\omega = 0.7 \, \mathrm{s^{-1}}$. **e** Simulated orientation (red) and velocity curves (black)

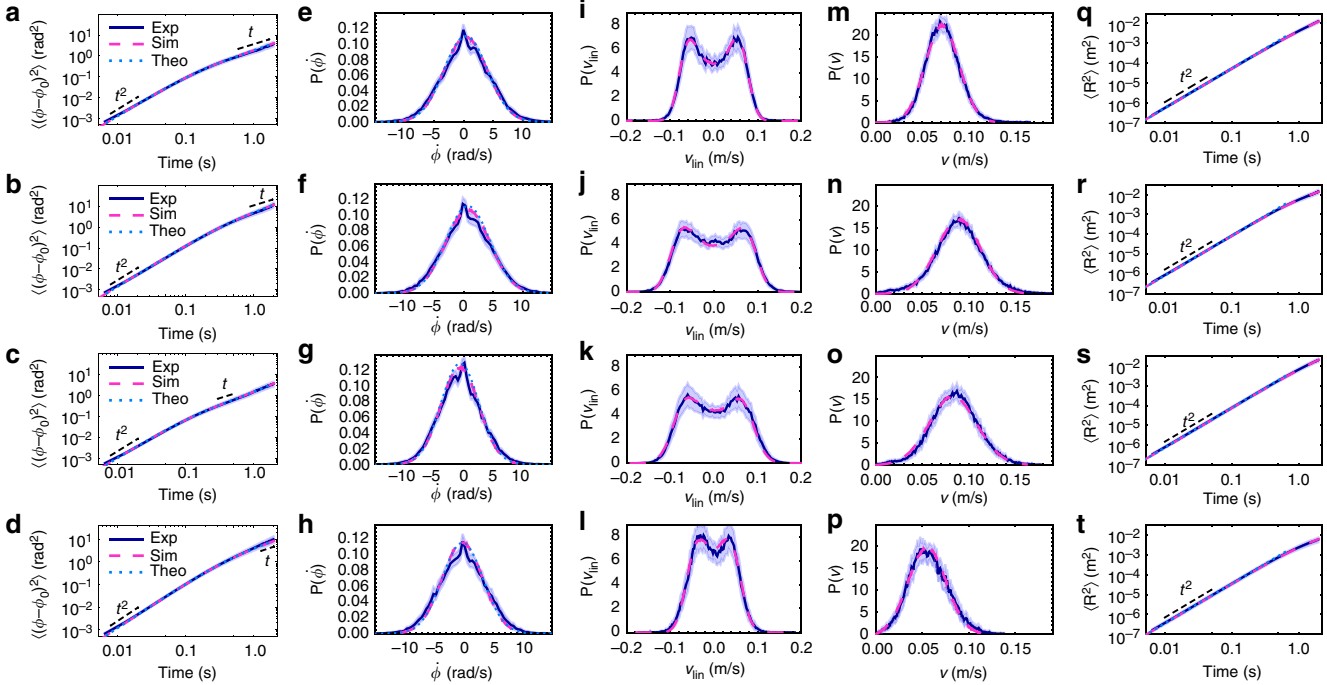

**Fig. 3** Determination of model parameters for the generic, carrier, tug and ring particles (ordered from top to bottom). **a–d** Rotational mean squared displacement, **e–h** rotational velocity distribution, **i–l** linear velocity distribution, **m–p** absolute velocity distribution, **q–t** translational mean squared displacement. Solid dark blue and dashed magenta curves show the experimental data and simulation results, respectively. Dotted light blue plots are the theoretical solutions. Experimental error intervals represent the standard error of the mean (3 s.e.m). The parameter values are listed in Supplementary Table 1

(Fig. 3i–p) and the translational MSDs (Fig. 3q–t), which can be directly fitted by Eqs. (4) and (5). The linear velocity distribution is not a simple Gaussian, but shows a double peak related to the activity. The absolute velocity distribution also clearly deviates from the two-dimensional Maxwell−Boltzmann distribution of passive particles, especially, the maximum is shifted by the propulsion force. The translational MSD mainly depicts the ballistic short-time behaviour, because the persistence length of our particles is of the order of the system size. To test the parameters on an independent quantity, we systematically compared the model with the measured inertial delay. We define the correlation function

$$C\big(\dot{\mathbf{R}}(t), \mathbf{n}(t)\big) = \big\langle \dot{\mathbf{R}}(t) \cdot \mathbf{n}(0) \big\rangle_{\mathrm{T}} - \big\langle \dot{\mathbf{R}}(0) \cdot \mathbf{n}(t) \big\rangle_{\mathrm{T}}, \quad (10)$$

i.e. the average difference between the projection of the orientation on the initial velocity and projection of the velocity on the initial orientation. This function starts at zero and re-approaches zero in the limit $t \to \infty$. In overdamped systems, Eq. (10) is zero at all times. In the underdamped case the velocity direction pursues the orientation and $C(\dot{\mathbf{R}}, \mathbf{n}(t))$ reaches its maximum after a specific delay. Pronounced peaks, related to the decay

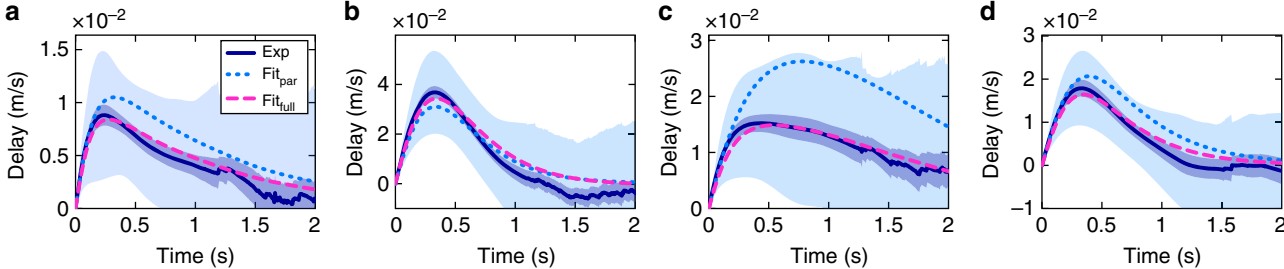

**Fig. 4** Time-dependence of the delay function. Delay functions for the **a** generic, **b** carrier, **c** tug and **d** ring particle. The solid dark blue curve shows the experimental result. The light blue area expresses the standard deviation of the ensemble, the dark blue area illustrates the standard error (2 s.e.m). The dotted light blue curve is the theoretical expectation for the parameters from Fig. 3. The dashed magenta line is the theoretical expectation where the parameters have been obtained from a full fit to velocity distributions, mean squared displacements and the delay function (see Supplementary Table 2)

numbers and $\tau_r$ are observed in Fig. 4a–d. The measurements and theoretical predictions using the parameters determined from Fig. 3 are consistent. Some deviations above the statistical error are visible, in particular for the tug particle (dotted line in Fig. 4c), due to overfitting the parameters, when the delay function is not explicitly taken into account. This is confirmed by a more general fit, which minimises the total mean squared error for the curves from Figs. 3 and 4. All curves obtained from this agree with the measurements within the statistical error (see dashed magenta lines in Fig. 4a–d).

**Inertial dependence**. Strikingly, both the short- and long-time particle dynamics in our system depend on the delay number $\mathfrak{D}_0$. The fundamental reason is the additional orientational correlation in Eq. (3), which is delayed by the rotational friction rate $\tau_r^{-1}$. The exponent in this expression represents the MSD of $\phi$, which is dominated by order $t^2$ at short times and order $t$ at long times. Consequently, neglecting external torque, this function follows a Gaussian decay at short times and an exponential decay at long times. The significance of the inertial delay is quantified by $\mathfrak{D}_0$. For small $\mathfrak{D}_0$, the correlation approaches the overdamped result and for large $\mathfrak{D}_0$ the correlation time is significantly delayed by $\tau_r$. To confirm this prediction, we compare the measured correlation functions and the solutions of Eq. (3). The results are consistent, as shown in Fig. 5a.

The numerical and analytical dependence of the ballistic and diffusive regimes on the moment of inertia are displayed in Fig. 5b, c, which show that $\langle \dot{\mathbf{R}}^2 \rangle$ and $D_L$ increase with $J$. The effects of finite $J$ can be simply demonstrated mathematically by expanding Eqs. (5) and (8) in the limit $J \to 0, \infty$. As $J$ vanishes, we find that

$$\lim_{J \to 0} \langle (\mathbf{R}(t) - \mathbf{R}_0)^2 \rangle = \left( 2D\frac{\xi}{M} + V_p^2 \frac{\xi}{\xi + MD_r} \right) t^2, \quad (11)$$

which agrees with the results reported in ref. [34]. For infinitely large $J$ we obtain

$$\lim_{J \to \infty} \langle (\mathbf{R}(t) - \mathbf{R}_0)^2 \rangle = \left( 2D\frac{\xi}{M} + V_p^2 \right) t^2, \quad (12)$$

which simply corresponds to the sum of the thermal and injected kinetic energies. For the long-time diffusion constant the asymptotic behaviour for small moments of inertia is

$$D_L = D + \frac{V_p^2}{2D_r} + \frac{V_p^2}{2\xi_r} J + \mathcal{O}(J^2), \quad (13)$$

which intuitively demonstrates, how, the leading order $J$ increases the persistence time (namely by a linear term proportional to ($\xi_r/$

$J)^{-1}$). The dependence of $D_L$ on $\mathfrak{D}_0$ has no upper bound, and its asymptotic behaviour is described by

$$D_L = D + V_p^2 \sqrt{\frac{\pi}{8D_r\xi_r}}\sqrt{J} + \mathcal{O}\left(\sqrt{J^{-1}}\right). \quad (14)$$

The origin of this dependence can be intuitively understood by considering the turn-around manoeuvre of a simple noise-free active particle. When a torque is applied perpendicularly to the velocity, the particle will turn around at point $P$ and eventually approach circular motion. As the moment of inertia quantifies the resistance of a particle to changing its angular momentum, a particle with low $J$ will turn faster than one with high $J$, as shown in Fig. 5d. This applies only to the transient states, where $\ddot{\phi} \neq 0$. In the steady state, the radius $r$ of the final circle is independent of $J$. The angular momentum of an active particle with random reorientations is constantly changing. Its inertia resists these changes and modifies the distribution of reorientations directly opposing the effect of rotational noise.

**Discussion**
Our observations demonstrate the profound influence of inertia on the long- and short-time dynamics of self-propelled particles. Considering the relevance of inertia[30], our model is applicable to various systems, such as levitating[38,39] and floating[40] granular particles and dusty plasmas[41]. It is straightforward to extend the model to elongated particles[5,11,12,14,37,42] and it was shown numerically that collective motion of rod-like particles is well described by similar equations of motion[37]. Qualitatively, in our system, rod-like particles show an inertial delay as well (see Supplementary Fig. 2). In a more general framework, diffusion and friction coefficient could be tensorial and additional non-linear force terms, such as a self-aligning torque reported in refs. [27,33], might be added to the force balance. Our model predicts that microswimmers perform a short-time ballistic motion like passive particles, but in practice, their motion also depends on their specific propulsion mechanism[43,44] and hydrodynamic effects[45,46]. Generally, the inertial effects will depend on the corresponding time scales in the system. In numerical experiments, this can be demonstrated by gradually reducing the density of hypothetical particles, retaining all other parameters as constants. At very low densities, the MSD exhibits four different regimes: short-time ballistic, short-time diffusive, active ballistic and long-time diffusive regime (see Fig. 6).

The long-time diffusion coefficient of passive particles is independent of inertia and is related to the friction coefficient via the Stokes−Einstein relation. However, for actively moving particles we find an explicit dependence on the moment of inertia

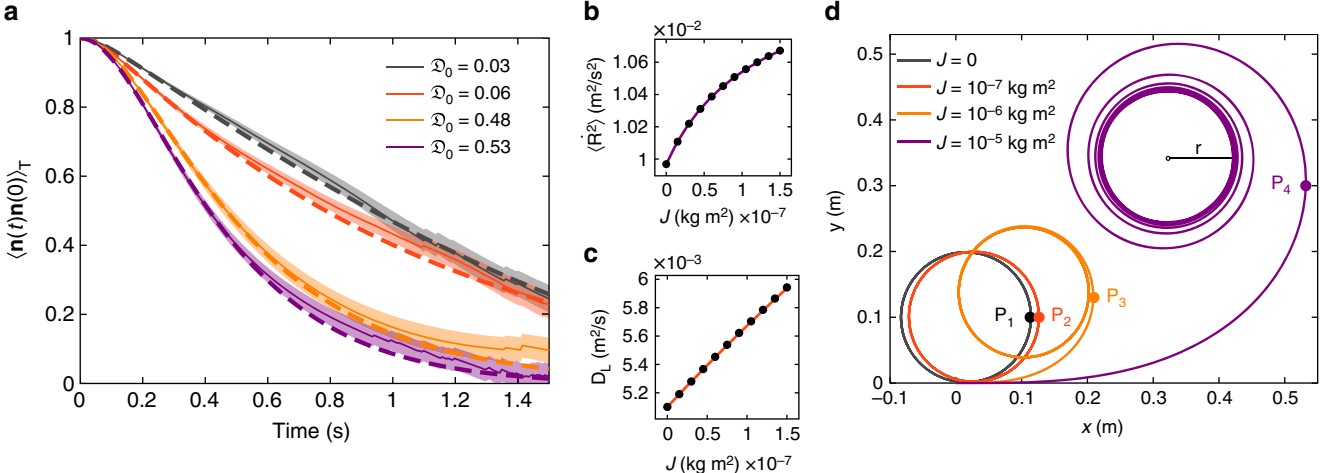

**Fig. 5** Particle dynamics dependence on rotational inertial delay. **a** Time dependence of orientational correlation functions. Solid lines represent the measurements, error bands represent the standard error of the mean. Dashed lines are the analytic results using the parameters from Fig. 3. **b** Slope of the ballistic regime, i.e. the second moment of the velocity $\langle \dot{\mathbf{R}} \rangle$, as a function of $J$ (the circles and solid line are the numerical results and the analytic solution to Eq. (5), respectively). The model parameters (except $J$) are those used in Fig. 6. **c** Long-time diffusion coefficient $D_L$ as a function of $J$ (the circles and solid line are the numerical results and the analytic solution to Eq. (8), respectively). **d** Trajectories of active particles under a constant torque applied at $t_0$. As $J$ increases the turn-around manoeuvre becomes increasingly difficult, so the distance and time increase until the turning point $P_{1,2,3,4}$ is reached

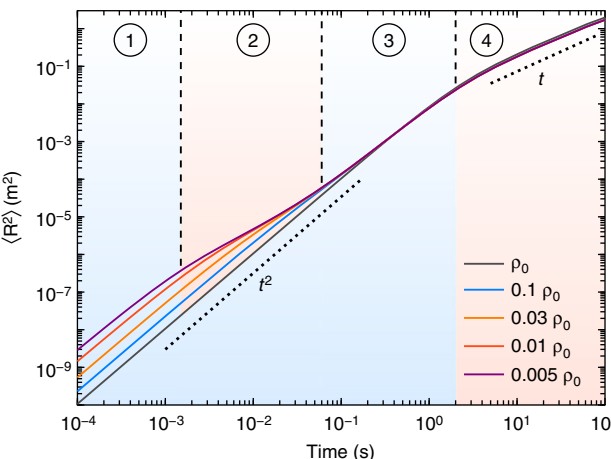

**Fig. 6** Time dependence of MSD for gradually decreasing density. MSD of hypothetical particles with successively reduced material density (i.e. $M$ and $J$ are reduced by a common fraction). The other parameters are fixed as $M_0 = 4$ g, $J_0 = 1.5 \times 10^{-7}$ kg m$^2$, $\xi/M = 6.5$ s$^{-1}$, $\xi_r/J = 5.5$ s$^{-1}$ $D = 1 \times 10^{-4}$ m$^2$ s$^{-1}$, $D_r = 1$ s$^{-1}$, $V_p = 0.1$ ms$^{-1}$, $\omega = 0$. When $\rho = \rho_0$, two regimes are visible. When the density drops below $0.01\rho_0$ the MSD divides into four regimes: 1. inertial ballistic, 2. short-time diffusive, 3. active ballistic and 4. active diffusive. In the limit of large damping and vanishing torque the 1–2, 2–3 and 3–4 transition times are given by $M/\xi$, $D/2V_p^2$ and $D_r^{-1}$, respectively

(with no explicit dependence on the total mass $M$). This finding illustrates the importance of $J$ for macroscopic self-propelled particles. While mass distribution and shape are generally important for efficient motion of animals[47–50] and adaptation to the environment[51,52], our results suggest that $J$ can be exploited in novel control strategies for active matter. Biological organisms cannot rapidly vary their mass, but they can change $J$ by moving their limbs. For instance, cheetahs use tail motion to stabilise fast turns[53]. While the effect on the long-time diffusivity of vibrobots is a few percent, our theory predicts that for flying and floating particles these changes are more dramatic. For similar sized

particles flying in air (e.g. insects) we can expect that friction is about two orders of magnitude smaller. Also, biological organisms can vary their moment of inertia dynamically for up to almost two orders of magnitude, depending on the position and the axis of rotation[54]. In this case, from Eq. (8), the long-time diffusion coefficient changes up to a factor of about three per order of magnitude in $J$. By increasing $J$, animals can then faster explore a large area. Conversely, by decreasing $J$, they can more easily dodge obstacles or predators and counteract sensorial[55] and behavioural delay[16]. Even under conditions, where animals cannot control their rotational deflections, such as aerodynamic turbulence, or during random collisions with neighbours[56], they could stabilise their movements through variations of $J$.

## Methods

**Particle fabrication.** Four particle types were designed and printed: The generic particle consists of a cylindrical core (diameter 9 mm, length 4 mm) topped by a cylindrical cap (diameter 15 mm, length 2 mm). Beneath the cap, seven tilted cylindrical legs (each of diameter 0.8 mm) were attached in parallel in a regular heptagon around the core. The legs lift the bottom of the body by 1 mm above the surface. The typical mass was about $m = 0.76$ g. From the mass and shape of the particle the moment of inertia was approximated as $J = 1.64 \times 10^{-8}$ kg m$^2$. To vary the propulsion velocity of the particles, we printed five types with different leg inclination angles 0°, 2°, 4°, 6° and 8°.

The carrier particle was fabricated with the same core as generic, but its cap was topped with a 1 mm tall, 8.5 mm diameter cylinder. The carrier socket held two galvanised steel washers, each with an outer diameter of 16 mm and a mass of 1.6 g. The leg inclination of carrier particles was fixed at 2°, and mass and moment of inertia were $m = 4.07$ g, $J = 1.46 \times 10^{-7}$ kg m$^2$, respectively.

The tug particle was a generic with a fixed leg inclination of 2° and thinner core (diameter 4 mm). This core held a hexagonal M5 threaded galvanised steel nut with a short diagonal and height of 8 and 3.75 mm, respectively. The mass and moment of inertia were $m = 1.57$ g and $J = 2.54 \times 10^{-8}$ kg m$^2$, respectively.

The ring particle had a leg inclination of 4° and a ring-shaped cap with a hole (diameter 9 mm) in the middle. The mass and moment of inertia were $m = 0.33$ g and $J = 1.26 \times 10^{-8}$ kg m$^2$, respectively.

All particles were labelled with a simple high contrast image allowing the detection software to identify the particle's position and orientation. The particles were printed from a proprietary methacrylate-based photopolymer (FormLabs Grey V3, FLGPGR03) of typical density 1.11(1) kg/L at a precision of 0.05 mm. They were subsequently cleaned in high purity (>97%) isopropyl alcohol in a still bath, followed by an ultrasound bath, then hardened by three 10-min bursts under four 9 W UVA bulbs. Finally, irregularities were manually filed away and the label sticker was attached.

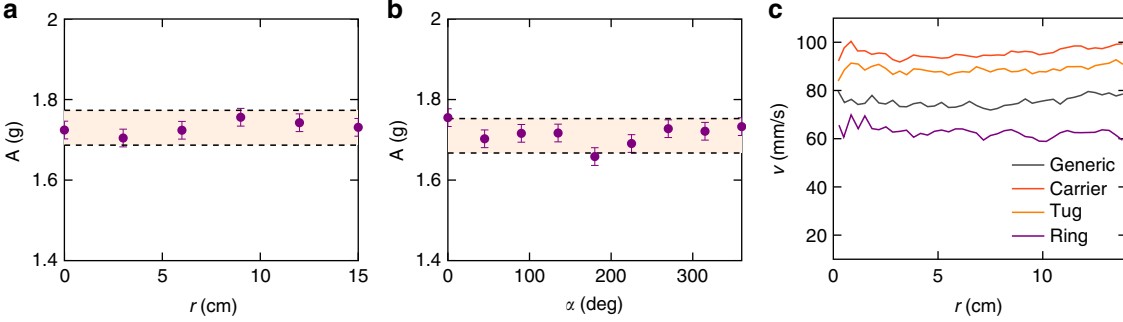

**Fig. 7** Spatial variation of excitation amplitude and particle velocity. Excitation amplitude in dependence of **a** radial positions at a fixed azimuthal angle ($\alpha = 0$) and **b** azimuthal angle at fixed radial position ($r = 6$ cm). The shaded area illustrates a width of ±2.5% around the mean. Error bars illustrate the sensorial accuracy (s.e.m.). **c** Average velocity of particles as a function of the radial position for all particle types

**Experimental setup**. The vibrobots were excited by vertical vibrations generated by a circular acrylic baseplate (diameter 300 mm, thickness 15 mm) attached to an electromagnetic shaker (Tira TV 51140) and surrounded by a barrier to confine the particles. The tilt of the plate was adjusted with an accuracy of 0.01°. The vibration frequency and amplitude was set to $f = 80$ Hz and $A = 66(4)$ μm, respectively, guaranteeing stable excitation with peak accelerations of 1.7(1) $g$ (measured by four LIS3DH accelerometers). To ensure homogeneous excitation, the acceleration amplitude was measured at different radial and azimuthal positions in steps of 3 cm and 45° respectively, at constant frequency. The variation of amplitudes at a mean acceleration of 1.7 $g$ is below 5% (see Fig. 7a, b). To ensure that no other factors significantly affect the isotropy of the system, the average particle velocity was measured as a function of the radial distance to the centre (Fig. 7c). The resulting fluctuations are small compared to the mean (standard deviation lies between 1.8 and 3.6% of the respective mean). Experiments were recorded using a high-speed camera system (Allied Vision Mako-U130B) operating at up to 152 fps with a spatial resolution of 1024 × 1024 pixels. Single particles were tracked to sub-pixel accuracy using standard image recognition methods. The tracking accuracy was determined from test measurements of a particle rigidly attached to the plate at different locations. A bivariate Gaussian distribution was fitted to the positions, from which the covariance matrix was obtained. The accuracy $2 \cdot \sigma_{max}$ is defined from the maximum of the diagonal entries in this covariance matrices $\sigma_{max}^2$ (see Fig. 8a). For the angular position, the error is directly obtained from the 95.4% confidence interval, since the distribution is non-normal due to pixel locking effects (see Fig. 8b). The resulting accuracy is ±4.7×10⁻⁴ m and ±0.013 rad. Multiple single trajectories were recorded for each particle, until 10 min of data were acquired. Events involving particle-border collisions were discarded.

**Analytic results**. The rotational behaviour of the particle was obtained by stochastic integration[57] of Eq. (2). The angular frequency and angular coordinate we obtained as

$$\dot{\phi}(t) = \omega + (\dot{\varphi}_0 - \omega)e^{-\xi_r t/J} + \sqrt{2D_r}\frac{\xi_r}{J}e^{-\xi_r t/J}\int_0^t dt' e^{\xi_r t'/J}\tau_{st}(t'), \quad (15)$$

and

$$\phi(t) = \varphi_0 + \omega t + \frac{\omega - \dot{\varphi}_0}{\xi_r}J(e^{-\xi_r t/J} - 1) + \sqrt{2D_r}\frac{\xi_r}{J} \\ \times \int_0^t dt' e^{-\xi_r t'/J}\int_0^{t'} dt'' e^{\xi_r t''/J}\tau_{st}(t''), \quad (16)$$

respectively. Here, $\phi_0$ and $\dot{\varphi}_0$ are initial angle and angular velocity, respectively, and the initial time was set to zero. As $\dot{\phi}(t)$ and $\phi(t)$ are both linear combinations of Gaussian variables, the corresponding probability distributions are also Gaussian. Thus, by calculating the mean

$$\langle\phi(t)\rangle = \varphi_0 + \omega t + \frac{\omega - \dot{\varphi}}{\xi_r}J(e^{-\xi_r t/J} - 1), \quad (17)$$

and the variance

$$\mu(t) = 2D_r t + \frac{2D_r}{\xi_r}J\left(e^{-\xi_r t/J} - 1 - \frac{(e^{-\xi_r t/J} - 1)^2}{2}\right), \quad (18)$$

one obtains the angular probability distribution

$$P(\phi, t) = \frac{1}{\sqrt{2\pi\mu(t)}}\exp\left(\frac{-(\phi - \langle\phi(t)\rangle)^2}{2\mu(t)}\right). \quad (19)$$

At times much longer than the reorientation time scale $1/D_r$ and the rotational

friction rate $J/\xi_r$, the variance of the angular distribution far exceeds $2\pi$, while the mean cycles between 0 and $2\pi$. This behaviour converges to the stationary state with a uniform distribution of $\phi$. At times much longer than the rotational friction rate $J/\xi_r$, the stationary distribution of the angular velocity reduces to

$$P(\dot{\phi}) = \sqrt{\frac{J}{2\pi D_r\xi_r}}\exp\left(\frac{-J(\dot{\phi} - \omega)^2}{2D_r\xi_r}\right). \quad (20)$$

The width of this distribution is inversely proportional to the moment of inertia.

From the translational equation of motion i.e. Eq. (1), the velocity in the laboratory frame of reference is obtained as

$$\dot{\mathbf{R}}(t) = \dot{\mathbf{R}}_0 e^{-\xi t/M} + \frac{\xi}{M}V_p e^{-\xi t/M}\int_0^t dt' e^{\xi t'/M}\mathbf{n}(t') \\ + \sqrt{2D}\frac{\xi}{M}e^{-\xi t/M}\int_0^t dt' e^{\xi t'/M}\mathbf{f}_{st}(t'), \quad (21)$$

where the initial velocity is denoted by $\dot{\mathbf{R}}_0$. The centre-of-mass position of a particle beginning its motion from the origin is calculated as

$$\mathbf{R}(t) = \mathbf{R}_0 + \dot{\mathbf{R}}_0\frac{M}{\xi}\left(1 - e^{-\xi t/M}\right) + \frac{\xi}{M}V_p\int_0^t dt' e^{-\xi t'/M} \\ \times \int_0^{t'} dt'' e^{\xi t''/M}\mathbf{n}(t'') + \sqrt{2D}\frac{\xi}{M}\int_0^t dt' e^{-\xi t'/M} \\ \times \int_0^{t'} dt'' e^{\xi t''/M}\mathbf{f}_{st}(t''). \quad (22)$$

The mean square displacement $\langle\mathbf{R}^2\rangle$ is obtained in the following integral form:

$$\langle\mathbf{R}^2(t)\rangle = \dot{\mathbf{R}}_0^2\frac{M^2}{\xi^2}\left(1 - e^{-\xi t/M}\right)^2 + 2V_p\left(1 - e^{-\xi t/M}\right) \\ \times \int_0^t dt' e^{-\xi t'/M}\int_0^{t'} dt'' e^{\xi t''/M}\dot{\mathbf{R}}_0\cdot\langle\mathbf{n}(t'')\rangle + \frac{\xi^2}{M^2}V_p^2 \\ \times \int_0^t dt' e^{-\xi t'/M}\int_0^{t'} dt'' e^{\xi t''/M}\int_0^t d\tau' e^{-\xi\tau'/M} \\ \times \int_0^{\tau'} d\tau'' e^{\xi\tau''/M}\langle\mathbf{n}(t'')\cdot\mathbf{n}(\tau'')\rangle + 4Dt \\ + \frac{4D}{\xi}M\left(e^{-\xi t/M} - 1 - \frac{1}{2}\left(e^{-\xi t/M} - 1\right)^2\right), \quad (23)$$

where $\langle\mathbf{n}(t)\rangle = e^{-\mu(t)/2}\left(\cos\langle\phi(t)\rangle, \sin\langle\phi(t)\rangle\right)$ and $\langle\mathbf{n}(t_1)\cdot\mathbf{n}(t_2)\rangle$ is defined by

$$\langle\mathbf{n}(t_1)\cdot\mathbf{n}(t_2)\rangle = e^{-D_r|t_1-t_2|}e^{D_r J/\xi_r}\exp\left[\frac{-D_r}{\xi_r}J \\ \times \left(e^{-\frac{\xi_r}{J}|t_1-t_2|} + e^{-\frac{\xi_r}{J}(t_1+t_2)} - \frac{1}{2}\left(e^{-2\frac{\xi_r}{J}t_1} + e^{-2\frac{\xi_r}{J}t_2}\right)\right)\right] \\ \times \cos\left[\omega(t_1 - t_2) + \frac{\omega - \dot{\phi}_0}{\xi_r}J\left(e^{-\frac{\xi_r}{J}t_1} - e^{-\frac{\xi_r}{J}t_2}\right)\right]. \quad (24)$$

The inertial delay correlation function Eq. (10) is given by

$$\langle\dot{\mathbf{R}}(t)\cdot\mathbf{n}(0)\rangle_T - \langle\dot{\mathbf{R}}(0)\cdot\mathbf{n}(t)\rangle_T = V_p\mathfrak{D}_2 e^{\mathfrak{D}_0}\mathfrak{D}_0^{(\mathfrak{D}_2-\mathfrak{D}_0)}e^{-t/\tau} \\ \times \text{Re}\left[\mathfrak{D}_0^{i\mathfrak{D}_1}\left(\mathfrak{D}_0^{-2\mathfrak{D}_2}\gamma(\mathfrak{D}_0 - i\mathfrak{D}_1 + \mathfrak{D}_2, \mathfrak{D}_0)\right.\right. \\ \left. -e^{2t/\tau}\mathfrak{D}_0^{-2\mathfrak{D}_2}\gamma(\mathfrak{D}_0 - i\mathfrak{D}_1 + \mathfrak{D}_2, \mathfrak{D}_0 e^{-t/\tau_r})\right. \\ \left. -\gamma(\mathfrak{D}_0 - i\mathfrak{D}_1 - \mathfrak{D}_2, \mathfrak{D}_0 e^{-t/\tau_r})\right. \\ \left.\left. +\gamma(\mathfrak{D}_0 - i\mathfrak{D}_1 - \mathfrak{D}_2, \mathfrak{D}_0)\right)\right]. \quad (25)$$

In the stationary case the Fokker−Planck equation of our model projected into one dimension reduces to

$$\int_{-\infty}^{\infty} d\phi\, \partial_{\dot{X}}\left(\frac{\xi}{M}V_p\cos\phi - \frac{\xi}{M}\dot{X} - D\left(\frac{\xi}{M}\right)^2\partial_{\dot{X}}\right)P(\dot{X}, \phi) = 0. \quad (26)$$

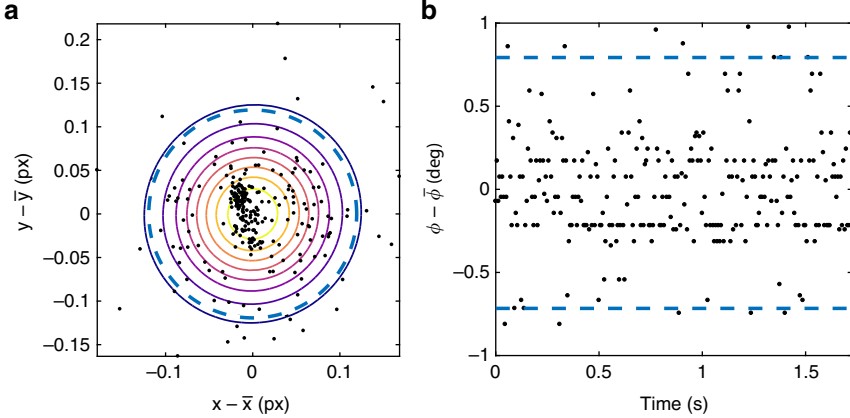

**Fig. 8** Accuracy of spatial and angular detection. **a** Tracked positions of a sticking particle during a time interval of 1.2 s. Solid lines show the isocontours of a bivariate mixed Gaussian distribution. The dashed line shows a circle with a radius equal to two times the square root of the maximum diagonal entry of the covariance matrix. **b** Orientations of a sticking particle. Dashed lines mark the confidence interval that includes at least 95.4% of the data points

One can anticipate the linear velocity distribution to be in the following form:

$$P(\dot{X}) = \frac{1}{\sqrt{2\pi q}} \int_{-\pi}^{\pi} d\phi \frac{1}{2\pi} \exp\left(-\frac{(\dot{X} - W\cos\phi)^2}{2q}\right), \quad (27)$$

where $q$ and $W$ are functions of $D$, $D_r$, $\tau$, $\tau_r$, $V_p$ and $\omega$. Analytic approximations for $q$ and $W$ are obtained from comparing the second and fourth moments to the solution of the Langevin equation(see Supplementary Methods).

**Parameter matching**. All parameters are obtained from fitting analytic and numeric results to the measurements of MSDs, velocity distributions and eventually the delay function, which describes a cross-correlation between orientation and velocity. The velocity in experiments is defined from the displacement of successive positions of the particle $\mathbf{v}(t) = (\mathbf{r}(t + \Delta t) - \mathbf{r}(t))/\Delta t$, where $\Delta t$ is the the time between two frames. Correspondingly the angular velocity is defined as $\dot{\phi} = (\phi(t + \Delta t) - \phi(t))/\Delta t$. The recording frame rate is 152 Hz, which corresponds to a minimal $\Delta t_0 = 0.0066$ s. When $\Delta t$ is small enough, i.e. below $\tau$, such that the ballistic motion of the particle can be captured accurately, the distribution of $\mathbf{v}$ and $\dot{\phi}$ will approach the stationary state. In our experiment we find $\tau$ is on the order of 0.1 s and $\Delta t = \Delta t_0$ is noticeably smaller. However, to ensure that the choice of $\Delta t$ does not significantly alter the parameters, fits with $\Delta t = 2, 3, 4\Delta t_0$ were checked and show no significant difference within the error bars of the parameters. The distribution and delay functions are provided for $\Delta t = 1, 2, 3, 4\Delta t_0$ in Supplementary Fig. 1 as reference. Note that for the experimental linear velocity distribution, i.e. the distribution of the components of the velocity vector, each trajectory is numerically rotated by a random angle to reduce anisotropy of the distribution that arises from the initial conditions, where each particle, at start, points towards the plate centre.

Initial parameters can be obtained from analytic results of the model directly. The parameters $\tau_r$, $D_r$ and $\tau_0$ are straightforwardly determined from fitting the well-known solution to Eq. (2) (Ornstein−Uhlenbeck process). The first moment of the angular velocity distribution gives $\tau_0 = \omega \tau_r$. Angular diffusion coefficient and relaxation time are determined from the fit to the angular MSD.

The determination of the remaining parameters $D$, $\tau$ and $V_p$ is more sophisticated. The analytic solution for the initial slope of the translational MSD is given by Eq. (5) and an analytic approximation of the linear velocity distribution is obtained from Eq. (27). The function $\mathfrak{f}(\mathfrak{D}_0, \mathfrak{D}_1, \mathfrak{D}_2)$ in Eq. (5) starts from zero and goes asymptotically to 1 as $\mathfrak{D}_2$ grows large, such that it is confined to the interval [0, 1]. This gives upper and lower bounds for $V_p$, namely

$$V_p \in \left[ \sqrt{\langle \dot{\mathbf{R}}^2 \rangle - 2D\tau^{-1}}, \sqrt{\frac{\langle \dot{\mathbf{R}}^2 \rangle - 2D\tau^{-1}}{\mathfrak{f}}} \right].$$ Accordingly, the following iterative

procedure is used to determine a set of parameters.

The iteration starts with the initial guess $V_p = \sqrt{\langle \dot{\mathbf{R}}^2 \rangle}$ and $\mathfrak{f} = 1$. The analytic approximation of the linear velocity distribution, Eq. (27), is fitted to the measurement to estimate $\tau$ and $D$. After this initial stage, there are two different choices for the post-iterations; either keeping $\mathfrak{f} = 1$ and changing the value of $V_p$ to $\sqrt{\langle \dot{\mathbf{R}}^2 \rangle - 2D\tau^{-1}}$, or computing the value of $\mathfrak{f}$ with respect to the estimated value for $\tau$ from the pre-iteration, such that $V_p = \sqrt{\frac{\langle \dot{\mathbf{R}}^2 \rangle - 2D\tau^{-1}}{\mathfrak{f}}}$. This leads to two estimates for $\tau$ and $D$ via fitting the linear velocity distribution to the outcomes of the experiment.

By comparing the agreement between the resulting $\tau$ and $D$ from both choices and the measurement through taking MSD and absolute velocity distribution into account, the estimate with the worst agreement gets discarded. The iteration continues with the accepted estimate for the set of values of the parameters until the resulting curves agree below the standard error.

The resulting set of parameters fit the experimental curves in Fig. 3 with high accuracy. Note that the delay function is not explicitly fitted in this scheme, but used as a cross-check of our parameters. However, this can potentially overfit the parameters, such as for the tug particle (see Fig. 4c). We additionally implemented a numerical optimisation routine (Nelder−Mead optimisation[58]), which fits the numerical solution of the model to all experiment curves (MSDs, velocity distributions and delay function, where velocities are defined such that they match the experimental time scale $\Delta t_0$), by minimising the weighted sum of the mean squared errors. For the generic, carrier and ring particle only minor improvements were found. In the case of the tug particle a significantly better agreement for the delay function can be found by slightly sacrificing the agreement of the other curves. In particular, $\tau$ and $D$ are sensitive to small variations. This is in accordance with our model, where only the product of $\tau$ and $D$ enters in dominating terms. Nevertheless, the deviation between parameters without and with taking the delay function into account vary in the worst case by about 50% (tug particle) in $\tau$ and $D$ and much less for all other parameters. Both sets of parameter values are shown in Supplementary Tables 1 and 2 for comparison. In the latter case an error estimate was obtained from the parameter variation that quadruples the mean squared error.

**Code availability**. The custom code that supports the findings of this study is available from the corresponding authors upon reasonable request.

## Data availability
The data that support the findings of this study are available from the corresponding authors upon reasonable request.

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

## Acknowledgements

We acknowledge funding by the German Research Foundation (Grant Nos. SCHO 1700/1-1 and LO 418/23-1).

## Author contributions

C.S. designed the experimental setup. C.S. and A.L. carried out the experiments. C.S., S.J., and A.L. analysed the measurements. S.J. and C.S. wrote the simulation code. S.J. and C.S. performed and analysed the simulations. S.J. developed the theoretical results. All authors discussed the results and wrote the manuscript.

## Additional information

**Competing interests:** The authors declare no competing interests.

