## [Peer Review File · Nature Communications]

Reviewers' comments:

Reviewer #1 (Remarks to the Author):

This paper discusses the effects of inertia on macroscopic particles propelled by the vibrations of a plate. These particles have already been used by the authors of this paper and similar particles with different variants have been used by other authors. The authors dub these particles 'vibrobots', others use different terms.

In the present paper, the authors show that such propelled particles have inherent inertial effects. This is not new as other authors have already noted and modeled such particles by keeping inertial effects. The main observation here is the consistent delay between velocity direction and orientation of these bots. This is where the paper becomes interesting since the authors use an underdamped Langevin model (where inertial terms are kept) for the translational and rotational motions of these particles to explain the dynamics of the particles and the observed delay. Particles with different masses and a priori moment of inertia are used and tracked. By solving the Langevin equations numerically and analytically and using appropriate approximations or rather asymptotic solutions, the authors show that the phenomenology of their vibrobots can be reproduced quantitatively allowing for the determination of all the parameters entering into play. There is quite a few parameters controlling this dynamics such as the different frictional coefficients which are a priori unknown and the nature and amplitude of the noise. By doing this analysis, the authors show how to recover these parameters from the tracking of the translational and rotational dynamics of the particles as well as their correlations.

This paper has the merit of clearly explaining the role of inertial effects in the mobility of such particles and gives analytic expressions that can be used to characterize similar particles from simply analyzing their tracks.

The results presented can therefore be valuable to other experimental systems and point to strategies for controlling the mobility of particles with inertia notably through controlling their moment of inertia.

The paper is very clearly written and the conclusions neatly supported by experiments and modeling. I am therefore inclined to recommend this paper for publication.

However, I have several points that I would like the authors to clarify before accepting the paper.

1) It is said in the paper (page 3) that the imperfections of the plate and some bouncing ball instability induce long time random motion. I do not fully understand the bouncing ball instability in the context of these bots and do not see how inhomogeneities in the plate arise in the model. I am guessing that such effects can arise both in the frictional effects and in the random forces or torques but can the authors explain such effects better and can these effects be tested (a rough plate and a smooth plate for example ?)

2) The bots used mostly differ by their mass but their shape remains roughly the same (cylindrical). In order to test the role of the moment of inertia shouldn't the authors change the shape of the particles (rod like for example) in a more convincing manner ?

3) Figure 4 is an essential figure of this paper. It seems that experiments, theory and simulations agree. However, the experimental errors are very large. I do not understand why. Since noise is present in simulations, are there errors in simulations (what is the standard deviation for the simulations ? Is this a matter of statistics ?

4) The authors use particles propelled by the vibrations of a plate. Other bots are self propelled through the use of embarked vibration systems and are rod like. Should we expect the delay between velocity and orientation to follow similar rules for such particles and does shape matter in the conclusions of this paper ? The authors should comment on the generality of these results to other real systems

5) The results obtained here pertain to a single particle trajectory and particle shape. In principle one

is interested in active particles because of their potential collective behavior. Are there implications for such behavior of the results obtained here? What possible implications for particle collisions can be expected from the delay characterized here or the variation of particle properties with moment of inertia? I am thinking of work by Dauchot and coll. where collisions between circular particles end up aligning the particles or work by Deblais et al. where rod like particles do not align due to inertia.

6) The authors state in the conclusion of the paper that strategies based on changes of moment of inertia may be devised following these results. I see in Fig. 5 that the velocity and the long time diffusion constant change by very little (a few percent) upon large changes in the moment of inertia. In Fig. 6, large variations of the mass are required to change the behavior of the MSD of hypothetical particles. The authors should be more explicit on how such small changes may be exploited to control mobility and what implications changes in the overall shape of the MSD (presence of additional regimes) has on the ability of these particles to control their mobility. This is an essential point of the paper so the authors need to be more precise on what they mean and eventually quantify these changes (the effects seem small from what they show).

Reviewer #2 (Remarks to the Author):

The authors study the effect of inertia on the dynamics of self-propelled particles. This is an important topic, which has been overlooked in the past years. Indeed most experimental systems dealt with small organisms such as bacteria or micron-sized particles such as Janus colloidal particles. It is however clear that the motion of more massive objects, such as large animals or vibrated grains, is dominated by inertial effects.

The paper is clear and well written and the data are rather convincing too. I, however, have a number of comments and questions, which I would like the authors to address before publication.

1) The model :

It was shown in ref [33] that, for a very similar system, the torque felt by the particles actually depends on the angle between the particle orientation and the velocity. It was later demonstrated that this ingredient is essential to the emergence of a spontaneous alignment of particles during collisions, which in turn can promote collective motion [27]. It is thus an important aspect of the dynamics, which is likely to be present here also. The analysis of the short time dynamics analysis should provide a way to clarify this. In the presence of this coupling torque, not only the velocity orientation turns toward the particle orientation, but also the other way round. Evidence should be provided that it is not the case here to justify the use of this simplified model to describe the experimental data.

2) Experimental set up :

The authors should provide some characterization of the homogeneity of the vibration. This could be obtained from the signals of the four accelerometers exploring different radial and azimuthal location on the vibrating plate.

What is the resolution on the measurement of the particle position?

3) Data analysis :

How is the velocity defined? The experimental data are intrinsically discretized, hence leading to the measurement of a displacement on a certain timescale. How are the results affected by the choice of this timescale?

I understand that the linear velocity is the projection of the velocity on the particle orientation. Since the velocity, or rather the displacement is computed from a time interval, the time at which the orientation is considered should be provided too. Is it averaged on the same time interval?

The authors should provide more details on the procedure followed to extract the parameters τ , D and V_p , especially given that the system size is smaller than the persistence length of the trajectory.

4) Minor comments:

There is a typo in the legend of fig 3: *expeirmental* instead of *experimental*, as well as later in the text, just before formula (14): *demonstrates* instead of *demonstrates*

In fig 4: I understand that both J and M are varied by tuning the density. From that point of view, the legend in the figure is a bit confusing: one may think that only the mass is varied. Also, how is J related to the mass in this numerical illustration?

REPLY to referee 1:

This paper discusses the effects of inertia on macroscopic particles propelled by the vibrations of a plate. These particles have already been used by the authors of this paper and similar particles with different variants have been used by other authors. The authors dub these particles vibrobots, others use different terms. In the present paper, the authors show that such propelled particles have inherent inertial effects. This is not new as other authors have already noted and modeled such particles by keeping inertial effects. The main observation here is the consistent delay between velocity direction and orientation of these bots. This is where the paper becomes interesting since the authors use an underdamped Langevin model (where inertial terms are kept) for the translational and rotational motions of these particles to explain the dynamics of the particles and the observed delay. Particles with different masses and a priori moment of inertia are used and tracked. By solving the Langevin equations numerically and analytically and using appropriate approximations or rather asymptotic solutions, the authors show that the phenomenology of their vibrobots can be reproduced quantitatively allowing for the determination of all the parameters entering into play. There is quite a few parameters controlling this dynamics such as the different frictional coefficients which are a priori unknown and the nature and amplitude of the noise. By doing this analysis, the authors show how to recover these parameters from the tracking of the translational and rotational dynamics of the particles as well as their correlations. This paper has the merit of clearly explaining the role of inertial effects in the mobility of such particles and gives analytic expressions that can be used to characterize similar particles from simply analyzing their tracks. The results presented can therefore be valuable to other experimental systems and point to strategies for controlling the mobility of particles with inertia notably through controlling their moment of inertia. The paper is very clearly written and the conclusions neatly supported by experiments and modeling. I am therefore inclined to recommend this paper for publication. However, I have several points that I would like the authors to clarify before accepting the paper.

Our reply: We thank the referee for the helpful recommendations and provide our point by point reply in the following. Changes in the manuscript are highlighted in blue colour.

It is said in the paper (page 3) that the imperfections of the plate and some bouncing ball instability induce long time random motion. I do not fully understand the bouncing ball instability in the context of these bots and do not see how inhomogeneities in the plate arise in the model. I am guessing that such effects can arise both in the frictional effects and in the random forces or torques but can the authors explain such effects better and can these effects be tested (a rough plate and a smooth plate for example ?)

Our reply: We agree that our explanation was incomplete. Following the referees comment, we have extended this paragraph (see p.3-4) to explain the background in more detail. In our vibrobot experiments we typically observe a significant randomness of motion, even on very smooth plates and purely periodic (sinusoidal) excitation. We investigated this in a previous study in great detail [New J. Phys. 12,123001 2016], where we found that at a sufficient excitation amplitude the particle will enter a tumbling regime, where all legs bounce off the plate asynchronously due to a transition into a bouncing ball instability. This asynchronous motion of the legs leads to a tiny and very irregular precession, which in turn leads to random reorientations of the particle.

The bots used mostly differ by their mass but their shape remains roughly the same (cylindrical). In order to test the role of the moment of inertia shouldn't the authors change the shape of the particles (rod like for example) in a more convincing manner ?

Figure R1 Inertial delay of a rod-like particle. (a) 3D-printed rod-like vibrobot with aspect ratio three, printed from clear resin. (b) Sample trajectory showing delay of velocity direction (black) and orientation angle (red). (c) Delay between velocity direction angle Θ (black) and orientation angle ϕ (red) demonstrates that Θ typically lags behind ϕ . (d) Delay function showing the characteristic shape.

Our reply: It is true that our particles are rotationally symmetric, but we change the moment of inertia by moving mass to the outer edge or the inside of the cylinder, which leads to significant changes of J up to one order of magnitude. The reason for remaining cylindrical shape is to minimize the effect of the shape asymmetry on the model parameters, in particular anisotropy of the diffusion and friction coefficients. While rod-like particles are undoubtedly interesting, in particular for collective motion, their rotational behaviour on the single particle level will still be described by a single degree of freedom in 2D (see also [Parisi et al. Sci. Rep. 8, 9133 (2018)]), but the translational equation of motion could become tensorial to include motion parallel and perpendicular to the orientation. This would increase the number of free parameters in the model further. We believe in the context of our paper it is better to keep the shape circular symmetric to avoid this. However, we have performed a trial measurement of a rod-like particle and qualitatively find the same behaviour concerning the inertial delay, as shown in Fig. R1. In particular we observe the delay between orientation and velocity, when the particle takes a sharp turn and observe the characteristic shape of the delay curve. We believe that for a quantitative description, the model for anisotropic particles would feature more general, tensorial diffusion constants and friction coefficients (which is part of our ongoing research). To emphasize this, we have added a corresponding paragraph to the discussion in the paper (see p.12 - 13), where we mention the relevance of our effect for rod-like particles.

Figure 4 is an essential figure of this paper. It seems that experiments, theory and simulations agree. However, the experimental errors are very large. I do not understand why. Since noise is present in simulations, are there errors in simulations (what is the standard deviation for the simulations? Is this a matter of statistics?)

Our reply: Following the referee's comments we believe that our presentation of Figure 4 was misleading. The experimental 'error bars' in the previous version depicted the standard deviation of the ensemble of trajectories, i.e. the differences between the delay functions measured for each realisation. Indeed this does not properly quantify the statistical error. A better quantifier is the standard error of the mean, which we plot in the revised version. Since we find a statistically significant deviation for the tug particle, we additionally performed a more in-depth parameter estimation to explain this. We performed a full fit of

the model which minimizes the weighted sum of the squared differences of MSDs, velocity distributions and delay functions using a downhill-simplex algorithm (Nelder-Mead method). This way we find a much better agreement for the delay function on the order of the standard errors (with only insignificant deviations from the previous curves in figure 3). This demonstrates that the first parameter determination from only the MSDs and velocity distributions overfits the problem, such that larger deviations appear for the delay function. Additionally this allows us to determine a meaningful uncertainty for all parameters (the error is defined by the parameter variation which quadruples the squared error). We added a revised version of Figure 4 (see p.9), the parameters obtained from the full fit and explain the procedure in more detail in the methods section (see p.18-20).

The authors use particles propelled by the vibrations of a plate. Other bots are self propelled through the use of embarked vibration systems and are rod like. Should we expect the delay between velocity and orientation to follow similar rules for such particles and does shape matter in the conclusions of this paper ? The authors should comment on the generality of these results to other real systems

Our reply: Indeed, other experiments use vibrational exciters attached to elongated particle, such as the commercially available hexbug toys. Very similar equations of motion to ours have been successfully used to describe a system of such active rods numerically in [Parisi et al. Sci. Rep.8,9133 (2018)], so it is fair to assume similar effects should be observable for such particles. As mentioned above, we also manufactured rod-like particles using our 3D printer and demonstrate that the inertial delay can be observed likewise for such particles. A corresponding paragraph has been added to the discussion of our paper (see p.12).

The results obtained here pertain to a single particle trajectory and particle shape. In principle one is interested in active particles because of their potential collective behaviour Are there implications for such behaviour of the results obtained here ? What possible implications for particle collisions can be expected from the delay characterized here or the variation of particle properties with moment of inertia? I am thinking of work by Dauchot and coll. where collisions between circular particles end up aligning the particles or work by Deblais et al. where rod like particles do not align due to inertia.

Our reply: We believe the most basic implication on collective behaviour lies in the fact, that underdamped particles will be able to store energy after collisions with other particles in translational and rotational degrees of freedom. We believe this alone should influence the state diagram of active particles considerably and could possibly break up motility induces clusters more easily than in overdamped systems. As shown in the works mentioned by the referee and also in our work, inertia can have several significant effects on vibrationally driven active matter. However, in particular our analytic results predict, that dramatic effects should occur once the relaxation time is on the order of the persistence time of the activity, such as for floating particles or flying organisms. Please see our reply to the next point, where we further explain these implications.

The authors state in the conclusion of the paper that strategies based on changes of moment of inertia may be devised following these results. I see in Fig. 5 that the velocity and the long time diffusion constant change by very little (a few percent) upon large changes in the moment of inertia. In Fig. 6, large variations of the mass are required to change the behaviour of the MSD of hypothetical particles. The authors should be more explicit on how such small changes may be exploited to control mobility and what implications changes in the overall shape of the MSD (presence of additional regimes) has on the ability of these particles to control their mobility. This is an essential point of the paper so the authors need to be more precise on

Figure R2 Prediction of the long-time diffusion coefficient for a centimeter-sized particle in a low-friction environment, such as floating or flying in air. Parameters are identical to that in Figure 5 in the main manuscript, but translational and rotational friction coefficients are reduced by two orders of magnitude.

what they mean and eventually quantify these changes (the effects seem small from what they show).

Our reply: It is true that the effect on D_l for vibrobot particles is about a few percent. However, in general this effect can be much more dramatic, in particular when inertial terms become even more significant. The range of moment of inertias considered by us is realistic, since the moment of inertia for a point mass grows quadratically with the radius from the centre of rotation, such that shifting mass away from the centre can increase J easily by up to two orders of magnitude (see e.g. [Zatsiorsky, Kinetics of human motion (2002), p.293]). Additionally, for particles or organisms flying in air, the (laminar) friction coefficient can be two orders of magnitude smaller, further increasing the effect. Our model predicts that similar particles flying in air (e.g. insects) should be able to modify D_l by a factor of 3 per order of magnitude (see Fig. R2). The delay could be particularly important when interaction between organisms in a swarm additionally suffers from sensorial and behavioural delay, where inertia is important to predict movements of neighbouring particles. We have modified the corresponding paragraph in the paper (see p.13), to make this point more clear to the reader.

We wish to thank the referee again for the helpful report. We hope that after our revision the referee can fully recommend the manuscript for publication in Nature Communications.

REPLY to referee 2:

The authors study the effect of inertia on the dynamics of self-propelled particles. This is an important topic, which has been overlooked in the past years. Indeed most experimental systems dealt with small organisms such as bacteria or micron-sized particles such as Janus colloidal particles. It is however clear that the motion of more massive objects, such as large animals or vibrated grains, is dominated by inertial effects. The paper is clear and well written and the data are rather convincing too. I, however, have a number of comments and questions, which I would like the authors to address before publication.

Our reply: We thank the referee for the helpful recommendations and address the concerns in the following point by point reply. Changes in the manuscript are highlighted in blue colour

The model: It was shown in ref [33] that, for a very similar system, the torque felt by the particles actually depends on the angle between the particle orientation and the velocity. It was later demonstrated that this ingredient is essential to the emergence of a spontaneous alignment of particles during collisions, which in turn can promote collective motion [27]. It is thus an important aspect of the dynamics, which is likely to be present here also. The analysis of the short time dynamics analysis should provide a way to clarify this. In the presence of this coupling torque, not only the velocity orientation turns toward the particle orientation, but also the other way round. Evidence should be provided that it is not the case here to justify the use of this simplified model to describe the experimental data.

Our reply: We agree that in general additional forces can contribute to the motion, such as the self-aligning torque in the pioneering work mentioned by the referee. In ref [PRL 110, 208001 (2013)] the governing equation for the rotational velocity is chosen proportional (with proportionality constant ζ) to the sine of the relative angle between velocity and orientation (polarity) of the particle, to account for this effect. We checked the relevance of this self-aligning torque in our system by calculating both sides of the governing equation (eq (2) in [PRL 110, 208001 (2013)]) for each point in time and analyse the resulting values of ζ . The results are displayed as histograms over all datapoints in Fig. R3. We find very strong fluctuations of ζ , as opposed to the model from the reference, which assumes a constant value for ζ . The mean value $\bar{\zeta}$ is on the order of the standard error of our measurement and more than two orders of magnitude smaller than the standard deviation of its distribution (i.e. practically zero and even negative for the tug particle). We believe this suggests, that this force is sufficiently weak in our system and overshadowed by the other factors, especially, since all other quantities are very well described by our model. A reason might be the difference in the particle architecture of [PRL 110, 208001 (2013)] and our particles. We designed our particles as symmetric as possible (only the tilt of the legs breaks the symmetry), to avoid the need for a more complicated model. Nevertheless, in a more general framework this term might be considered, as mentioned by the referee. We have added a clarifying section to the discussion (see p.12), where we address this.

Experimental set up: The authors should provide some characterization of the homogeneity of the vibration. This could be obtained from the signals of the four accelerometers exploring different radial and azimuthal location on the vibrating plate. What is the resolution on the measurement of the particle position?

Our reply: Indeed, granular experiments are known to be sensitive to inhomogeneities of the plate vibrations, which needs to be addressed. We measured the acceleration amplitude at different radial positions and azimuthal positions in steps of 3 cm and 45 degrees respectively at the frequency and amplitude parameters used in the article. We find the variation of amplitude at a mean acceleration of 1.7 g is below

Figure R3 Distribution of proposed proportionality factor $\zeta = \omega/(\sin \alpha \text{sgn}(\cos \alpha))$ [PRL 110, 208001 (2013)] in our data for (a) generic, (b) carrier, (c) tug and (d) ring particle. The mean of the distribution $\bar{\zeta}$ is below the standard error and two orders of magnitude below the standard deviation of the distribution.

5% (with a standard deviation of about 0.03 g, but we assume a very conservative estimate for the variation of 3σ), comparable to previous examples from the literature. Since we also want to ensure this variation and other factors do not significantly affect the trajectories, we additionally calculated the average particle velocity as a function of the radial distance to the centre. Here we do *not* find a systematic variation above the statistical fluctuations. We added a new figure (Figure 7) and paragraph to the methods section, where we provide this information (see p.15-16).

Furthermore, the tracking accuracy of the measurement was determined from test measurements of a particle rigidly attached to the plate at different locations. For the position a bivariate Gaussian distribution is fitted to the tracking results, from which the covariance matrix is obtained. We use the maximum of the diagonal entries in the covariance matrices σ_{\max}^2 and define the error as $2 \sigma_{\max}$. For the angular position, we use the same principle but define the error from the 95.4% confidence interval, since the distribution is clearly not normal, due to pixel locking effects. This way we obtain an accuracy of about ± 0.15 px, or $\pm 4.7 \times 10^{-4}$ m (2σ) and ± 0.013 rad. This accuracy sets the lower limits of the y-axis in the MSD plots in figure 3. We added this information and an additional figure (Figure 8) to the corresponding paragraph in the methods section (see p.15-16).

Data analysis: How is the velocity defined? The experimental data are intrinsically discretized, hence leading to the measurement of a displacement on a certain timescale. How are the results affected by the choice of this timescale? I understand that the linear velocity is the projection of the velocity on the particle orientation. Since the velocity, or rather the displacement is computed from a time interval, the time at which the orientation is considered should be provided too. Is it averaged on the same time interval? The authors should provide more details on the procedure followed to extract the parameters τ , D and V_p , especially given that the system size is smaller than the persistence length of the trajectory.

Our reply: As mentioned by the referee, the velocity is defined from the displacement of successive positions of the particle $\mathbf{v}(t) = (\mathbf{r}(t + \tau_0) - \mathbf{r}(t))/\Delta t$, where Δt is the time between two frames. Our recording frame rate is 152 Hz, which provides a minimal Δt_0 of about 0.0066 seconds. For the particle motion in the long-time diffusive regime the distribution of \mathbf{v} measured in this fashion, will depend on the value of Δt and converge towards a sharp peak for large Δt that corresponds to the mean velocity. However, when Δt is small enough, such that the ballistic motion of the particle can be captured accurately, the distribution of \mathbf{v} will approach its stationary form. This is the case in our experiments and we do not observe significant changes of the velocity distribution, even when we increase Δt by a factor of four. We have added an extended section to the methods part (see p.19) and an additional supplementary figure to describe the

relevance of this and demonstrate the robustness of our method. In our case, the linear velocity refers not to the projection on the orientation, but the components of the velocity vector ($P(v_x) = P(v_y)$) parallel and perpendicular to an (arbitrary) lab frame. We clarify this in the revised version (see p.8). Additionally, we describe our parameter matching and fitting procedure in more detail. In response to a question of referee 1, we have also extended our analysis of the parameters in relation to the delay function and provide additional information on this (see p.18-20).

Minor comments: There is a typo in the legend of fig 3: *expeirmental* instead of *experimental*, as well as later in the text, just before formula (14): *demonstrates* instead of *demonstrates*. In fig 4: I understand that both J and M are varied by tuning the density. From that point of view, the legend in the figure is a bit confusing: one may think that only the mass is varied. Also, how is J related to the mass in this numerical illustration?

Our reply: We have implemented the corrections and modified the legend in figure 6 to state the density instead of the mass. J and M are proportional to each other in this simulation, assuming that indeed, only the density is gradually reduced, while the hypothetical ‘shape’ remains constant.

We wish to thank the referee again for the helpful report. We hope that after this revision the referee can fully recommend the manuscript for publication in Nature Communications.

REVIEWERS' COMMENTS:

Reviewer #1 (Remarks to the Author):

I have now read both the authors' reply to my comments and the changes brought to the text and find both satisfying. I have one final request, before fully accepting this manuscript, that the authors add their findings for a rod (figures included in the response) to the supplementary material. I understand that the problem is more complex for such a geometry but I think this data gives even more strength to what the authors are reporting for cylindrical particles.

Reviewer #2 (Remarks to the Author):

As I said in my previous report, studying the effect of inertia on the dynamics of self-propelled particles is an important topic, which has been overlooked in the past years. The paper is clear and well written and the data are rather convincing too.

The authors have seriously addressed the points I was making in my first report. I, therefore, recommend publication.

REPLY to referee 1:

I have now read both the authors' reply to my comments and the changes brought to the text and find both satisfying. I have one final request, before fully accepting this manuscript, that the authors add their findings for a rod (figures included in the response) to the supplementary material. I understand that the problem is more complex for such a geometry but I think this data gives even more strength to what the authors are reporting for cylindrical particles.

Our reply: We thank the referee for his suggestions and have added the figure on the rod particle to the supplementary information, as well as a remark in the main manuscript (see p. 12).